# The Impact of Crystal Light Yield Non-Proportionality on a Typical Calorimetric Space Experiment: Beam Test Measurements and Monte Carlo Simulations

**Lorenzo Pacini** [1,*], **Oscar Adriani** [1,2], **Eugenio Berti** [1,2], **Pietro Betti** [2], **Gabriele Bigongiari** [3,4], **Lorenzo Bonechi** [1], **Massimo Bongi** [1,2], **Sergio Bottai** [1], **Paolo Brogi** [3,4], **Guido Castellini** [1,5], **Caterina Checchia** [3,4], **Raffaello D'Alessandro** [1,2], **Sebastiano Detti** [1], **Noemi Finetti** [1,6], **Paolo Maestro** [3,4], **Pier Simone Marrocchesi** [3,4], **Nicola Mori** [1], **Miriam Olmi** [1,2], **Paolo Papini** [1], **Claudia Poggiali** [1,2], **Sergio Ricciarini** [1,5], **Piero Spillantini** [1,2], **Oleksandr Starodubtsev** [1], **Francesco Stolzi** [3,4], **Alessio Tiberio** [1] and **Elena Vannuccini** [1]

1    INFN Firenze, Via B. Rossi 1, I-50019 Firenze, Italy
2    Department of Physics and Astronomy, University of Florence, Via G. Sansone 1, I-50019 Firenze, Italy
3    Department of Physical Sciences, Earth and Environment, University of Siena, I-53100 Siena, Italy
4    INFN Pisa, Largo B. Pontecorvo, I-56127 Pisa, Italy
5    IFAC CNR, Via Madonna del Piano 10, I-50019 Firenze, Italy
6    Department of Physical and Chemical Sciences, University of L'Aquila, Via Vetoio, Coppito, I-67100 L'Aquila, Italy
*    Correspondence: lorenzo.pacini@fi.infn.it

**Abstract:** Calorimetric space experiments were employed for the direct measurements of cosmic-ray spectra above the TeV region. According to several theoretical models and recent measurements, relevant features in both electron and nucleus fluxes are expected. Unfortunately, sizable disagreements among the current results of different space calorimeters exist. In order to improve the accuracy of future experiments, it is fundamental to understand the reasons of these discrepancies, especially since they are not compatible with the quoted experimental errors. A few articles of different collaborations suggest that a systematic error of a few percentage points related to the energy-scale calibration could explain these differences. In this work, we analyze the impact of the nonproportionality of the light yield of scintillating crystals on the energy scale of typical calorimeters. Space calorimeters are usually calibrated by employing minimal ionizing particles (MIPs), e.g., nonshowering proton or helium nuclei, which feature different ionization density distributions with respect to particles included in showers. By using the experimental data obtained by the CaloCube collaboration and a minimalist model of the light yield as a function of the ionization density, several scintillating crystals (BGO, CsI(Tl), LYSO, YAP, YAG and BaF2) are characterized . Then, the response of a few crystals is implemented inside the Monte Carlo simulation of a space calorimeter to check the energy deposited by electromagnetic and hadronic showers. The results of this work show that the energy scale obtained by MIP calibration could be affected by sizable systematic errors if the nonproportionality of scintillation light is not properly taken into account.

**Keywords:** cosmic rays; calorimetry; scintillation; light yield

## 1. Introduction

Several relevant open questions regarding astroparticles and dark-matter physics require accurate measurements of cosmic rays (CRs). For instance, the direct observation of electron and positron spectra above a few TeV provides unique information regarding high-energy CR sources near Earth and dark-matter models, while CR acceleration and propagation models benefit from the accurate measurement of proton and nuclei spectra [1]. Space spectrometers such as PAMELA [2] and AMS-02 [3] are capable of separating matter and antimatter, but they cannot detect particles above a few TeV due to the limited acceptance and maximal detectable rigidity (MDR). Thus, the calorimetric technique is

employed to explore higher energies: a few examples of running space calorimeters are CALET [4] and DAMPE [5], while a future detector is HERD [6]. Even if the precision of recent direct CR measurements is strongly increased with respect to previous instruments, evidence of the disagreement among different experiments exists.

Figure 1 [7] shows the recent measurements of the electron and positron flux. Here, two groups of experiments are clearly present: from $\sim 100$ GeV to $\sim 1$ TeV DAMPE [8] and Fermi-LAT [9] feature higher fluxes with respect to those of CALET [10] and AMS-02 [11]. The differences are larger than the errors quoted by the experiments, which could be explained if unaccounted systematic errors are present. Besides electrons and positrons, other examples of tensions in CR measurements are the carbon, oxygen and iron spectra: CALET [12,13] results feature higher overall normalization with respect to the result of AMS-02 [14,15], even if the spectral shape is similar.

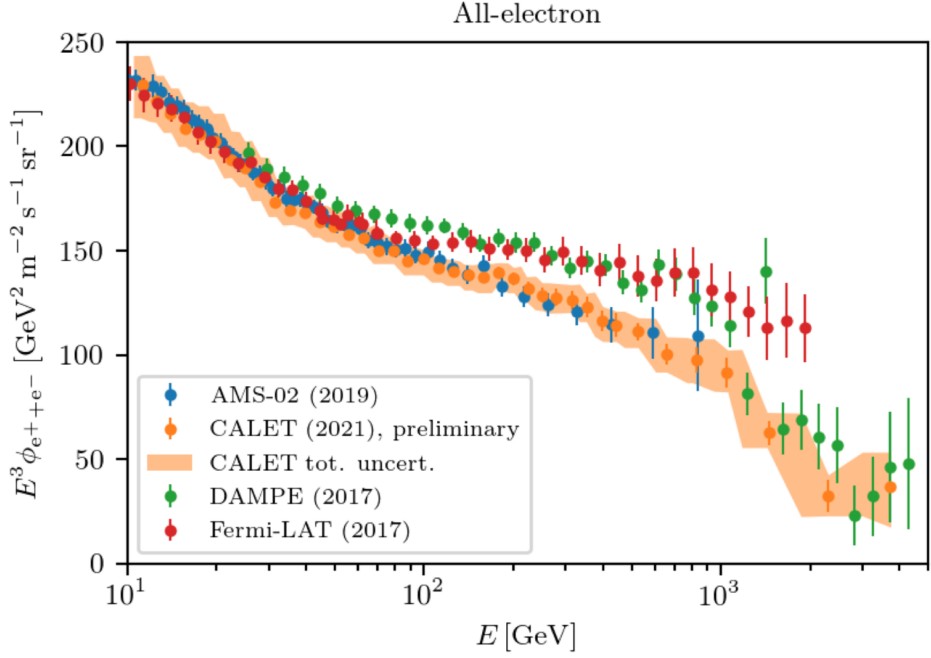

**Figure 1.** Recent direct measurements of the electron flux [7].

Since space calorimeters are usually composed of inorganic scintillators, the main goal of this work is to study the effect of the nonproportionality of the scintillation light yield for the measurement of high-energy showers, which could translate into a sizable systematic error on the energy scale of space experiments. A list of experiments employing inorganic crystals is shown in Table 1. The work presented in this contribution is also discussed in detail in [16].

**Table 1.** List of space cosmic-ray experiments based on calorimetric instruments produced with inorganic scintillators, and the main characteristics of the calorimeters.

| Experiment | Material | Electromagnetic Depth ($X_0$) | Hadronic Depth ($\lambda_I$) | Launch Year |
|---|---|---|---|---|
| CALET [4] | PWO | 27 | 1.2 | 2015 |
| DAMPE [5] | BGO | 32 | 1.6 | 2015 |
| FERMI [17] | CsI(Tl) | 8.6 | 0.4 | 2008 |
| HERD [18] | LYSO | 55 | 3.0 | 2027 (expected) |

## 2. Nonproportional Light Response: Minimalist Approach

The light response of both organic and inorganic scintillators depends on the ionization density [19], which is proportional to the energy deposit inside the crystals per unit length

($dE/dx$). In this paper, the Minimalist Approach model [20] was employed to approximate the nonproportionality of scintillators since it fit the data used to characterize the materials well (see Section 4). This model takes into account two effects. The first, the quenching [21] (or Birks) effect, dominates at high excitation density. A slightly modified version of this model was employed here: as proposed in [22], assuming a division of the energy deposition into cylindrical core and halo regions surrounding the particle trajectory, only the charge carriers inside the core are affected by the quenching effect. The relative light-emission efficiency formula for the modified Birk effect is then:

$$L_B = \frac{1 - \eta_H}{1 + B(1 - \eta_H) \times \frac{dE}{dx}} + \eta_H \tag{1}$$

where $B$ is the Birks parameter, and $\eta_H$ is the fraction of carriers in the halo region.

The second, called the Onsager effect, dominates at low excitation density and is described in [23]. Electrons and holes that initially do not form excitons can be recombined afterwards only if they are closer than the Onsager radius. This effect improves the efficiency of light emission, which can be written as follows:

$$L_O = 1 - \eta_{e/h} \exp\left(-\frac{(dE/dx)}{(dE/dx)_O}\right) \tag{2}$$

where $(dE/dx)_O$ is the strength of the Onsager term, and $\eta_{e/h}$ is the fraction of initial carriers that do not form excitons.

By combining Equations (1) and (2), the relative light-emission efficiency can be expressed as follows:

$$L = \left[1 - \eta_{e/h} \exp\left(-\frac{(dE/dx)}{(dE/dx)_O}\right)\right] \times \left[\frac{1 - \eta_H}{1 + B(1 - \eta_H) \times \frac{dE}{dx}} + \eta_H\right]. \tag{3}$$

### 3. Monte Carlo Simulation of the Ionization Density

To study the $dE/dx$ in different materials, a simulation code based on FLUKA [24] was employed. In order to improve the simulation accuracy, the minimal energy thresholds for particle tracking were set to be 1 keV for electrons and 100 eV for photons. Furthermore, all physical processes that contribute to ionization were activated. The output of the simulation is the amount of energy released for every bin of ionization density. Since the light signal depends on energy loss and light-emission efficiency, it can be computed as follows.

$$S_L = \sum_i \Delta E_i \times L_i. \tag{4}$$

where $S_L$ is the light signal in arbitrary units, and $\Delta E_i$ and $L_i$ are the energy loss and light-emission efficiency in a given bin of ionization density, respectively.

Two examples of the simulation output are shown in Figure 2. The mean energy deposit due to nonshowering protons (helium nuclei) crossing 2 cm of LYSO is shown in the black (red) histogram. These histograms also include the energy deposited by secondary particles (e.g., $\delta$ rays). As expected, the helium ionization density profile is different with respect to the one of protons, and it features the main peak at ∼24 MeV/cm, and a secondary peak at ∼6 MeV/cm, which is due to $\delta$-ray emission. Figure 2 also shows the typical light-emission efficiency [25] for alkali (green) and silicate (blue) scintillators (e.g., CsI(Tl) and LYSO). Even if the ratio between the mean energy deposit of helium nuclei and protons is 4, the ratio of the light signals is less (greater) than 4 for silicate (alkali) scintillators due to the different luminous efficiencies.

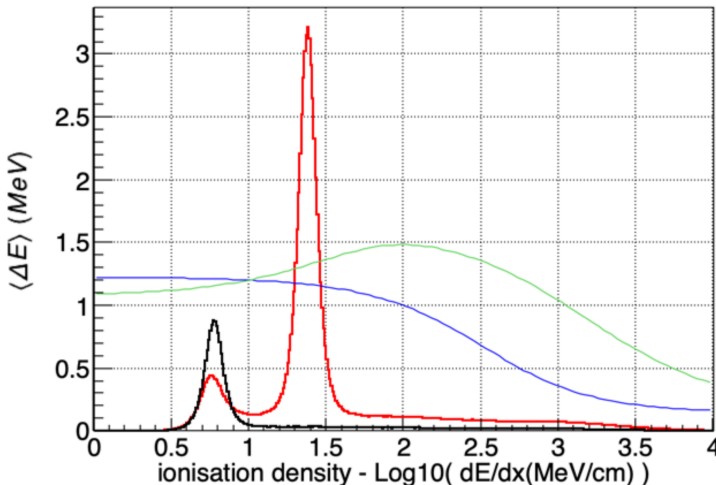

**Figure 2.** Black and red histograms represent the mean energy deposit for each bin of ionization density due to 100 GeV nonshowering protons and helium in 2 cm of LYSO, respectively. The green and blue curves are the typical luminous efficiencies for alkali and silicate scintillators, reported here in arbitrary units [16].

## 4. Characterization of Scintillators with CaloCube Beam Test Data

Typical methods employed to study the scintillator nonproportionality are based on Compton electrons and photon response, but in this work, the ionization produced by high energy nuclei is used; this technique was exploited by GLAST/Fermi-LAT [26] and DAMPE [27]. Different nonshowering nuclei feature different mean ionization densities that can be used to measure light signals corresponding to different ionization densities. By employing nuclei from protons ($Z = 1$) to argon ($Z = 18$), the light signals corresponding to ionization densities between 5 MeV/cm and 2 GeV/cm can be measured.

These measurements were performed with the CaloCube collaboration [28], which was an R & D project that exploited new concepts for the design of a space calorimeter. The main design is a 3D-segmented, homogeneous, isotropic, cubic calorimeter produced with cubic scintillating crystals. The acceptance of this instrument is larger than that of typical space telescopes since it measures particles coming from each side and not only from the zenith. The read-out system of the scintillating light consists of a pair of photodiodes with different active areas and double-gain custom front-end electronics. In order to optimize the design, different scintillating materials were tested with the Monte Carlo simulations of the calorimeter [29]. Furthermore, few CaloCube prototypes were built [30]. For instance, the performance of the large-scale prototype with high-energy electrons was described in [31]. In this work, data acquired with the prototype tested at the CERN SPS accelerator with high-energy nuclei in 2015 are discussed. This prototype consisted of several trays equipped with CsI(Tl) crystals, while the last tray allocated different cubic scintillators (test crystal), as shown in Figure 3.

The properties of the test crystals are summarized in Table 2.

The scintillating light was read out with a photodiode (VTH2090) coupled with custom front-end electronics, which mainly consisted of CASIS [32] chips.

During the test, the beam consisted of nucleus fragments with $30 \cdot A \, GeV$ kinetic energy, an $A/Z$ ratio equal to 2, and charge ranging from 1 to 18. Specific runs were employed to acquire events in which the beam directly hit each test crystal. During the data analysis, nonshowering nuclei were selected with the information of the nearby CsI crystals, while the impact position and the charge of the particle were reconstructed using a silicon tracker placed upstream of the CaloCube prototype.

The main results of this test are summarized in Figure 4, where different markers show the mean value of different crystal signals divided by the square of the nucleus charge. Considering an ideal scintillator featuring constant light-emission efficiency, the

points would be displaced on a horizontal line; for the tested scintillators, a clear deviation from this ideal condition was shown. Different trends of the point series are related to the different material nonproportionality of the scintillation light.

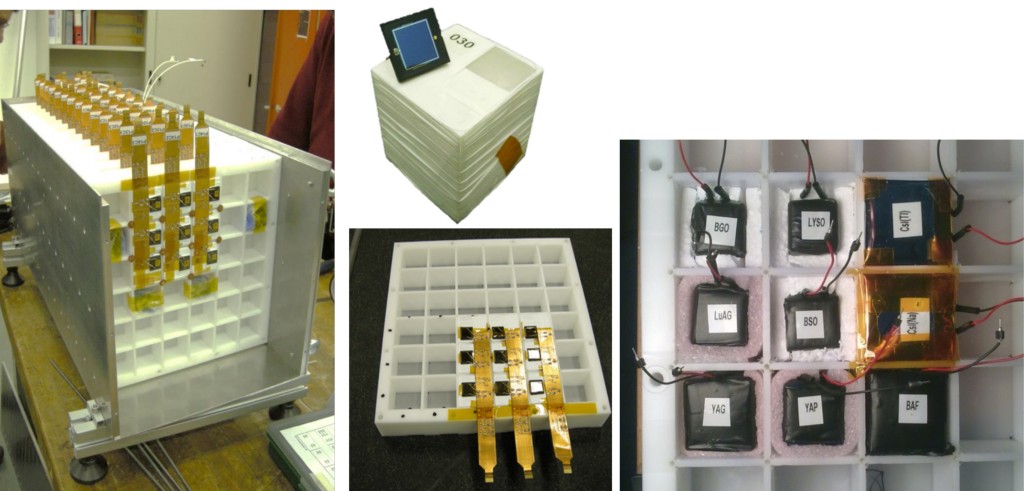

**Figure 3.** Left panel: image of the CaloCube prototype made of CsI crystals. Central panels: image of a prototype layer with CsI crystals (bottom panel) and a crystal with a VTH2090 PD (top panel). Right panel: image of the last tray, which includes different scintillators.

**Table 2.** Main properties of the tested materials, where $\rho$ is the density of the crystal, $\lambda_i$ and $X_0$ are the interaction and radiation lengths, respectively, $\lambda_{max}$ is the wavelength of the scintillation light at the emission maximum, and $\tau_{decay}$ is the decay time.

| Material | Size (cm) | $\rho$ (g/cm$^3$) | $\lambda_I$ (cm) | $X_0$ (cm) | $\lambda_{max}$ (nm) | $\tau_{decay}$ (ns) |
|---|---|---|---|---|---|---|
| BGO | 2.0 | 7.1 | 23 | 1.1 | 480 | 300 |
| CsI(Tl) | 3.6 | 4.5 | 40 | 1.9 | 550 | 1220 |
| LYSO | 2.0 | 7.4 | 21 | 1.1 | 420 | 40 |
| YAP | 2.2 | 5.5 | 22 | 2.7 | 370 | 27 |
| YAG | 2.5 | 4.6 | 25 | 3.5 | 550 | 70 |
| BaF$_2$ | 3.1 | 4.9 | 31 | 2.0 | 300 | 650 |

By exploiting the data shown in Figure 4, the relative light yield of each scintillator as a function of the nuclei charge was computed. For instance, the black points of Figure 5 show the relative light yield for CsI(Tl) and LYSO normalized to $Z = 18$. The minimalist approach was then used to fit the data by exploiting the $dE/dx$ profile obtained with the FLUKA simulation, as explained in the previous section; the red points of Figure 5 show the fit results. The simple model was able to reproduce the experimental trends of each crystal, reducing $\chi^2_{red}$ from 0.64 to 1.64. The fit results regarding the parameters of Equation (3) are summarized in Table 3.

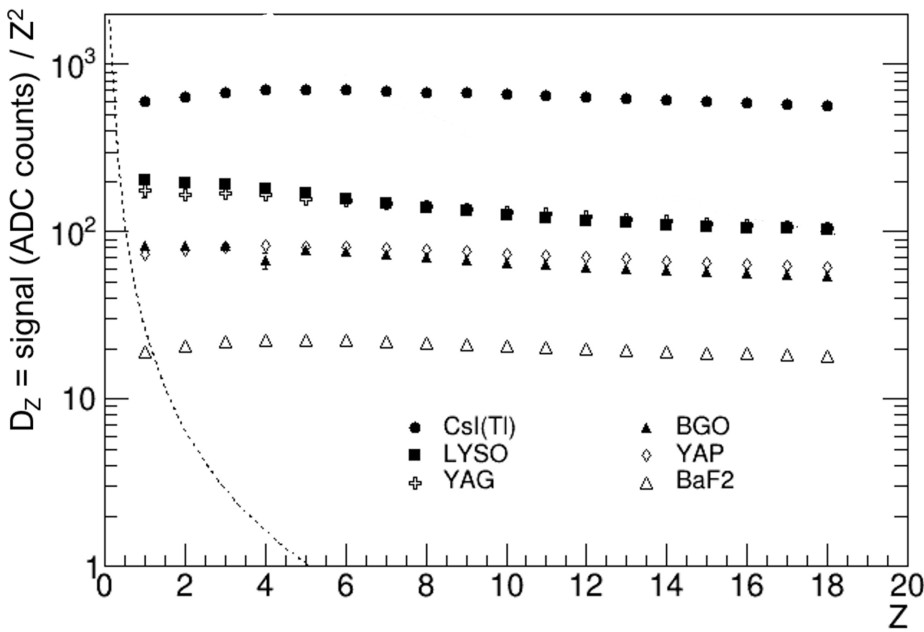

**Figure 4.** Different markers show the mean of the light signals obtained with different crystals crossed by different high-energy nuclei, divided by $Z^2$, and plotted as a function of Z. The dashed line represents the noise level [16].

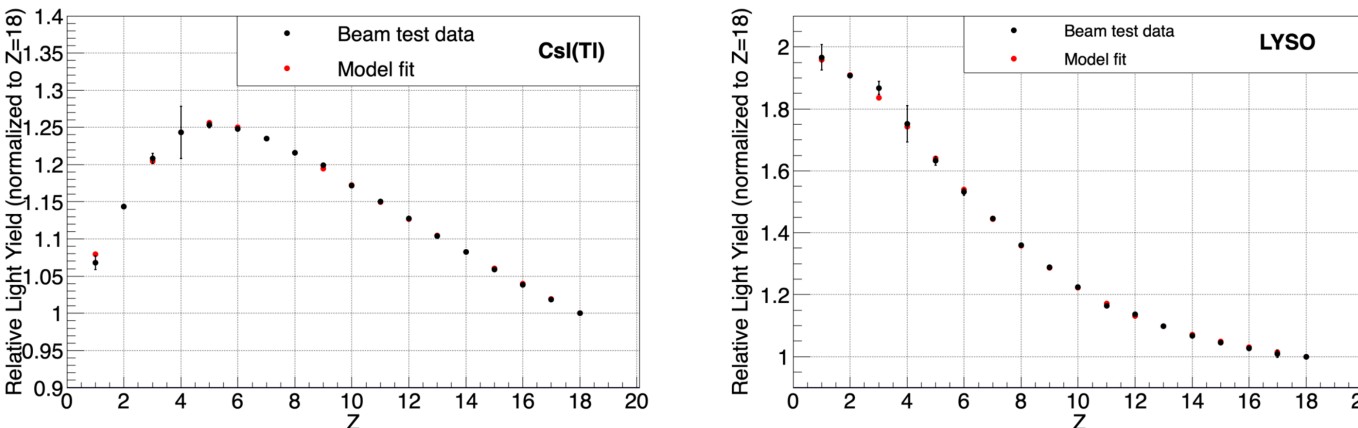

**Figure 5.** Relative light yield normalized to the argon one of (left panel) CsI(Tl) and (right panel) LYSO. Black points are the result of beam test data analysis, while red points are the result of the fit with the minimalist model; Equation (3) [16].

**Table 3.** Parameters of Equation (3) obtained with the fit of CaloCube data [16].

| Material | $\eta_{e/h}$ | $(dE/dx)_O$ MeV/cm | $\eta_H$ | (1/B) MeV/cm | $\chi^2_{red}$ |
|---|---|---|---|---|---|
| BGO | $0.159 \pm 0.033$ | $98 \pm 45$ | $0.1884 \pm 0.0039$ | $364 \pm 42$ | 1.64 |
| CsI(Tl) | $0.326 \pm 0.010$ | $34.1 \pm 2.8$ | $0.121 \pm 0.012$ | $1338 \pm 64$ | 0.81 |
| LYSO | $0.758 \pm 0.045$ | $164.7 \pm 8.4$ | $0.0274 \pm 0.0048$ | $45.1 \pm 9.1$ | 0.64 |
| YAP | $0.2212 \pm 0.0085$ | $90 \pm 11$ | $0.174 \pm 0.012$ | $873 \pm 70$ | 1.24 |
| YAG | $0.0912 \pm 0.015$ | $73 \pm 29$ | $0.1052 \pm 0.0055$ | $462 \pm 31$ | 1.23 |
| BaF$_2$ | $0.322 \pm 0.024$ | $35.8 \pm 6.2$ | $0.3440 \pm 0.0071$ | $546 \pm 36$ | 1.11 |

## 5. Simulation of a Space Calorimeter

The final section of this work is the study of a possible systematic error on the energy measurement obtained with a space calorimeters. A very simple geometric configuration,

i.e. a homogeneous cube of 1 m$^3$, is simulated with the FLUKA package: the active materials alternatively employed in the simulation are LYSO, BGO and CsI. In a real experiment, the effect of the nonproportionality of the light yield depends on several parameters e.g., the crystal manufacturer, the specific detector geometry and calibration, the front-end electronics and acquisition system. With the approach described in this contribution, the possible existence of systematic effects is discussed while quantitative results for specific running experiments can not be determined.

The typical calibration of space calorimeters involves several steps, e.g., the CALET calibration is described in [33]. Usually, one of the first steps makes use of the energy deposited by Minimum Ionizing Particles (MIP), i.e., nonshowering protons and helium nuclei. Thus, the calibration of the energy scale strongly relies on MIP measurements. Unfortunately, the ionization profile of the energy deposited by MIPs is different with respect to the one of an electromagnetic or hadronic shower. The simulation shows that even if the ionization profile of electron showers is different with respect one the MIP one, this is almost constant with the energy form 10 GeV to 1 TeV, while the profile of hadronic showers due to proton interactions depends on the energy. Figure 6 shows the ionization profile of electrons and protons obtained with the simulation and the light-emission efficiency of LYSO obtained with the fit described in previous section.

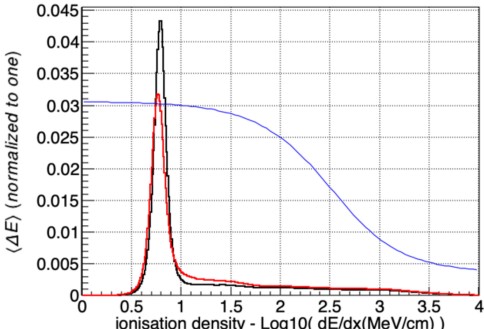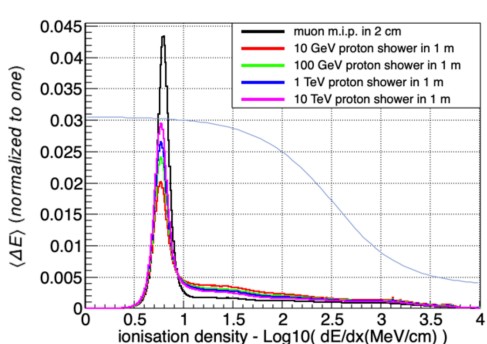

**Figure 6.** (**Left panel**) Ionization density profile of 10 GeV electrons (red line), MIP (black line), and the light-emission efficiency of LYSO (blue line) in an arbitrary unit. (**right panel**) Ionization density profile of protons at different energies (colored tick lines), MIP (black line), and the light-emission efficiency of LYSO (thin line) in an arbitrary unit [16].

As explained in Section 3, for a given energy deposit, a different ionization density profile translates into a different light output due to the nonproportional light-emission efficiency of the scintillator. For instance, a systematic shift of about 2.3% of the measured total energy deposit in LYSO crystals for electrons from 10 GeV to 1 TeV is obtained. Due to the steepness of cosmic-ray spectra, this energy shift translates into a sizable error on the normalization of the flux: assuming a spectral index equal to $\gamma$, and a systematic energy shift of $\Delta$, the shift of the flux normalization was $(\gamma - 1) \cdot \Delta$. The electron spectral index was $\sim$3; thus, the normalization shift obtained with the LYSO crystal was about $\sim$5%. Furthermore, the energy shifts obtained for protons depend on energy; thus, these affect both the shape and the normalization of reconstructed spectra.

The main results of this work, i.e., the systematic energy shift obtained with a 1 m$^3$ homogeneous calorimeter produced with different crystals, are summarized in Table 4.

**Table 4.** Energy shift due to the nonproportionality of light yield for a 1 m$^3$ homogeneous calorimeter produced with different scintillating materials [16].

| Scintillator | Electrons $\geq$10 GeV | Protons 10 GeV | Protons 100 GeV | Protons 1 TeV | Protons 10 TeV |
|---|---|---|---|---|---|
| LYSO | −2.3% | −7.1% | −5.6% | −4.6% | −3.4% |
| BGO | −1.1% | −4.3% | −3.0% | −2.3% | −1.8% |
| CsI(Tl) | +0.82% | +2.9% | +2.0% | +1.5% | +1.2% |

## 6. Conclusions

In this contribution, CaloCube data and the minimalist approach were employed to characterize the nonproportionality of scintillator light outputs. These results were then used to study the impact of this effect on homogeneous space calorimeters. Assuming that the energy scale of this kind of detector is mainly calibrated with MIP, an effect on the energy measurement of few percentage points exists. This leads to systematic shifts of the reconstructed spectral normalization of up to 5%. Regarding the results published with the running experiments, it is not clear if this effect had already been considered for the energy scale estimation since it was not mentioned in the main papers regarding calibration and energy reconstruction.

In order to accurately take into account the nonproportionality of scintillators, future calorimetric experiments might apply a similar procedure to the one presented in this contribution. The first step is the characterization of the material by employing a read-out system that is used during inflight operation, e.g., by using the high-energy nucleus technique described here. The second step could be to estimate the impact of nonproportionality on the energy scale and on other important parameters related to shower reconstruction, e.g., electron or proton discrimination variables. Eventually, if the effect is sizable, as suggested in this work, the last step could be the implementation of this effect inside the simulated data reconstruction software by using the ionization density profile and the material characterization.

**Author Contributions:** Conceptualization, P.P., E.V., O.A. and R.D.; software, N.M.; investigation, E.B., P.B. (Pietro Betti), L.P., O.S. and S.D.; data curation, E.V., M.O., C.P. and E.B.; writing—original draft preparation, L.P.; writing—review and editing, P.P. and E.V.; supervision, G.B., L.B., M.B., S.B., P.B. (Paolo Brogi), G.C., C.C., N.F., P.M., P.S.M., S.R., P.S., F.S. and A.T. All authors have read and agreed to the published version of the manuscript.

**Funding:** This research received no external funding.

**Data Availability Statement:** Not applicable.

**Conflicts of Interest:** The authors declare no conflict of interest.

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
