# Peer review of "The Impact of Crystal Light Yield Non-Proportionality on a Typical Calorimetric Space Experiment: Beam Test Measurements and Monte Carlo Simulations"

_instruments, doi:10.3390/instruments6040053_

Round 1

Reviewer 1 Report

The typesetting and english grammar can be improved. Examples: units are to be typeset in roman font, not math. Caption of figure 4 is unclear.

Following equation 1: \eta_H is the fraction of carriers ***in*** the halo, not outside. 

Figure 2 and 6: what are the units used for the efficiency curves?

Author Response

Dear referee,

thank you very much for your suggestions, which has been implemented in the new version of the article.

  • Caption 4 has been updated. This figure is quite complicated, since it represents the main result obtained with CaloCube data. For each different crystals, the mean light signal is measured. The signals are then divided by Z2 and plotted as a function of Z.

  • Equation 1: thank you very much for your correction. It was a error.

  • Figure 1 and 6: the efficiencies are in arbitrary units. We updated the figure captions with this information. Here the efficiencies are used only for comparison with the ionization density profiles.

Best regards,

Lorenzo Pacini

Reviewer 2 Report

Line 100, do you mean fig. 5 instead of fig. 4?

Line 101, chi^2 --> chi^2_{red}

Author Response

Dear referee,

thank you very much for your corrections, which has been implemented inside the new version.

  • "Line 100, do you mean fig. 5 instead of fig. 4?" Yes, it is a mistake

  • "Line 101, chi^2 --> chi^2_{red}." I updated the text.

Best regards,

Lorenzo Pacini

Reviewer 3 Report

Dear Editor and Authors,

the proceedings by Pacini et al. on the non-proportionality of light signals in space calorimeters is very interesting. I do have some comments/questions which I hope can be addressed prior to publication:

Physics/Methods:

you talk about non-interacting protons and other ions. This is however not the case - if these would be truly non-interacting they would also not ionise the media. I think you mean non-showering ...

Section 3 and Figure 2: It is not clear at what energies the protons and alpha particles are simulated. For the interpretation of Fig 4 the bin width and normalisation has to be given. I.e. the integral over the histogram should be the total deposited energy per particle on average. The bin-width on the x-axis should be given as parameter on the y-axis title. It's probably close to 0.01 ... The caption should indicate that the efficiencies are plotted in arbitrary units in the same frame. Could you elaborate more on the lower peak in the red spectrum? If the peak is from the electrons then the x-axis is not the dE/dx of alphas in this case but of electrons. This is not clear from the description in the caption nor the main text. It seems that secondary emissions are allowed to contribute to the observable on the x-axis and this should then be clearly stated. 

In L118-120 you write that EM showers have almost constant ionisation profiles between 10 GeV and 1 TeV while in L126 you say that the profile is changing and that it causes a 2.3% shift. Since this is one of the main results you need to explain better how the 2.3% are obtained and why it does not contradict the earlier statement. 

Figure 6: These 2 are apparently normalised differently than Figure 2. Yet here the actual normalisation is claimed to be to unity. If that would be the case, all areas under the curves should be identical. The black curves in the plots seem to cover larger areas than the colored ones. Could you please check that indeed the areas are identical? Also here the efficiencies are plotted in the same histogram without stating that their y-values are not represented by the y-axis and are arbitrarily scaled instead.

If I understand your conclusion correctly you claim that at least 2 of the experiments have not properly calibrated their instrument and hence get a wrong energy measurement. Why should this effect start at energies >100 GeV only? MIP energies for electrons should be in the MeV range - so if MIP calibration is the culprit as you say we should expect significant deviations already for 1 GeV electrons, no?

Language:

L34,37: presents -> present

L38: measurement -> measurements

Page 3 2nd sentence: the this -> this

Page 3 sentence below eq. (1): Birk' s -> Birks

L49: E_i -> Delta E_i

L72: particle -> particles

L75: simulation -> simulations

L80: of CsI -> CsI

L86: "test crystals" -> "test crystal"

Best regards,

A reviewer

Author Response

Dear referee,

thank you very much for your corrections, we also appreciated your comments and questions. Here you can find our comments:

  • “You talk about non-interacting protons and other ions…” Your comments is correct, but in several papers about direct measurements of cosmic rays “non-interacting” is used in order to indicate protons and nuclei when hadronic interactions does not occur. Of course, elector-magnetic interactions (e.g. ionization) are always present, while the likelihood of an hadronic interaction is small inside typical space calorimeters. So, usually “non-interacting” is a common sentence to indicate hadrons which does not produce showers inside calorimeters. Anyway, we changed “non-interacting” with “non-showering”.

  • “Section 3 and Figure 2: It is not clear at what energies the protons and alpha particles are simulated.” Particles are simulated at high energy (100 GeV), we added this information in the figure caption. Fortunately, the density profile of MIP does not depends on the energy in this energy region.

  • “For the interpretation of Fig 4 the bin width and normalisation has to be given….” As a general comment, since this is a proceeding related to a talk presented at CALOR2022 conference, which summarized an already published work, more information about the procedures included in this paper are in https://iopscience.iop.org/article/10.1088/1748-0221/17/08/P08014. Regarding this item, your understanding of the figure is correct. We did not think that a common reader would need information about the binning. We think that this work must be taken as a suggestion for future (and current) experiments while the quantitative results are only trivial estimations, as explained in section 6. So the reader should not focus on the detail of each step, while understanding the main procedure in order to avoid a big impact of this non-linearity on experimental systematic error. Eventually, each experiment should repeat this study by proper adjusting several parameter (including the binning, range of ionization density, ...) in order to match specific requirements of the experiments.

  • “The caption should indicate that the efficiencies are plotted in arbitrary units in the same frame.” thank you for the suggestion, we added this indication.

  • “Could you elaborate more on the….”. Alpha are particles which hit the crystal, but after the impact few delta-rays can be emitted inside the crystal. Those delta-rays (and other secondary particles, e.g. low energy photons) are generated inside the crystal due to the interaction of the primary alpha particle with the material itself. The energy deposit by secondaries must be included inside the ionization profile of alpha, in order to proper take into account all the processes contributing to the energy loss. We slightly modified the text to be more clear.

  • “In L118-120 you write that EM showers have almost constant ionisation profiles…” We tried to improve these sentences, here you can find an additional explanation: the shift obtained with electron showers with respect to the MIP is 2.3%. This value is almost the same for electrons from 10 GeV to 1 TeV. The ionization profile of electron is constant with the electron energy, even if it is different with respect the MIP profile, so the shift is constant but it is not equal to zero.

  • “Figure 6: These 2 are apparently normalised differently ….” We checked that the integrals (including the underflow and overflow bins) are the same for each curves. The black histogram is higher than the other ones, but it is also slightly narrower and the tail is lower than colored ones. We also added to the caption the indication of arbitrary unit for efficiencies. Here the efficiencies are used only for comparison with the ionization density profiles so we think that the reader only need the relative trend of those, while the absolute vale is not relevant.

  • “If I understand your conclusion correctly you claim..” As explained in the conclusion, we can not strongly claim about other experiment. We only notice that this effect is not mentioned in main papers regarding calibration and energy scale of space experiment. Furthermore, our work suggests that this effect is not negligible, even if the quantitative results can not be directly used, since each experiment should accurately study this effect within their framework (electronics, analysis and calibration procedure, etc…). Your point regarding the low energy region (< 100 GeV) is correct. We expect a energy shit, starting from hundreds MeV electrons. Unfortunately, we did not run simulation for such low energy, we could publish new results about this energy region if the scientific community is interested in.

Finally, thank you very much for the correction of the English, I’m sorry for the errors.

Best regards,

Lorenzo Pacini